# Clinical characteristics of COVID-19 patients hospitalized at Clinique Ngaliema, a public hospital in Kinshasa, in the Democratic Republic of Congo: A retrospective cohort study

**Junior Rika Matangila**[1,2]\*, **Ritha Kibambe Nyembu**[1,3], **Gloire Mosala Telo**[1], **Christian Djuba Ngoy**[1], **Taty Msueun Sakobo**[1], **Joel Mbembo Massolo**[1], **Blandine Mbo Muyembe**[1], **Richard Kapela Mvwala**[1], **Cedric Kabemba Ilunga**[1,4], **Emmanuel Bakilo Limbole**[1], **Jeff Mukengeshay Ntalaja**[1], **Roger Minga Kongo**[1]

1 Clinique Ngaliema, Kinshasa, République Démocratique du Congo, 2 Département de Médecine Tropicale, Université de Kinshasa, Kinshasa, République Démocratique du Congo, 3 Département de Biologie Médicale, Cliniques Universitaires de Kinshasa, Kinshasa, République Démocratique du Congo, 4 Département de Médecine Interne, Cliniques Universitaires de Kinshasa, Kinshasa, République Démocratique du Congo

\* matangilaj@yahoo.fr

**Data Availability Statement:** The data underlying this study cannot be shared publicly, as it is the

## Abstract

### Objectives

To describe the clinical characteristics of patients infected with SARS-CoV-2 at Clinique Ngaliema, a public hospital, in Kinshasa, in the Democratic Republic of Congo (DRC).

### Methods

This retrospective study analyzed medical records including socio-demographics, past medical history, clinical manifestation, comorbidities, laboratory data, treatment and disease outcome of 160 hospitalized COVID-19 patients, with confirmed result of SARS-CoV-2 viral infection.

### Results

The median age of patients was 54 years (IQR: 38–64), and there was no significant gender difference (51% of male). The most common comorbidities were hypertension (55 [34%]), diabetes (31 [19%]) and obesity (13 [8%]). Fever (93 [58%]), cough (92 [57%]), fatigue (87 [54%]), shortness of breath (72 [45%]) and myalgia (33 [21%]) were the most common symptoms, upon admission. Patients were categorized into mild (92 [57%]), moderate (19 [12%]) and severe (49 [31%]). Severe patients were older and were more likely to have comorbidities, compared to mild ones. The majority of patients (92% [147 of 160]) patients received hydroxychloroquine or chloroquine phosphate. Regression model revealed that older age, lower SpO2, higher heart rate and elevated AST at admission were all risk factors

---

property of the Clinique Ngaliema Hospital. For these reason, the hospital would prefer to share data for research purposes, upon reasonable request, accompanied by a research proposal with a detailed description of the objectives of the study, sent directly to the steering committee of Clinique Ngaliema Hospital at the following e-mail address: infor@cliniquengaliema.org. The authors did not receive any special privileges in accessing the data

**Funding:** The author(s) received no specific funding for this work.

**Competing interests:** The authors have declared that no competing interests exist.

associated with in-hospital death. The prevalence of COVID-19 and malaria co-infection was 0.63% and 70 (44%) of all patients received antimalarial treatment before hospitalization.

## Conclusion

Our findings indicated that the epidemiological and clinical feature of COVID-19 patients in Kinshasa are broadly similar to previous reports from other settings. Older age, lower SpO2, tachycardia, and elevated AST could help to identify patients at higher risk of death at an early stage of the illness. *Plasmodium spp* co-infection was not common in hospitalized COVID-19 patients.

## Introduction

The novel coronavirus, SARS-CoV-2 or 2019-nCoV (COVID-19), was first reported in a cluster of patients with pneumonia, in the Chinese city of Wuhan on 31 December 2019 [1]. Since then, the fast spread of the virus has resulted in a global pandemic that the WHO declared on 11[th] March 2020 [2]. At the beginning of the outbreak, China was the most affected country worldwide, from where the epidemic spread rapidly in Europe and America, with the USA being nowadays the most affected country. The Africa continent as well is affected by COVID-19, where many countries have weaker health systems and overall poorer responses to the virus. In this part of the planet, COVID-19 was expected to have higher attack and mortality rates [3].

While an increasing number of publications have brought significant insight on the clinical characteristics of infected patients from Asia [4, 5], Europe [6, 7] and America [8], there is a paucity of data on the clinical features of COVID-19 in sub-Saharan African population. In-fact recent data have shown disparities by race (black vs white) [9], income and access to health care [10] in regard to COVID-19 related severity and death, suggesting that the epidemiology, the clinical presentation and the course of COVID-19 may vary from one setting to another. Moreover, considering that malaria and COVID-19 share a couple of symptoms, identifying and managing COVID-19 cases may become challenging in sub-Saharan region. This underlines the necessity to provide sufficient data on COVID-19 characteristics in sub-Saharan African population, in order to guide medical practitioners and decision makers acting in this region.

This study describes the epidemiological and clinical features, laboratory findings, treatment, and outcomes of COVID-19 patients in one of the selected Hospital for COVID-19 care of Kinshasa, in DRC.

## Materials and methods

### Study design

This was a retrospective cohort study including patients of Clinique Ngaliema, one of the five public hospitals of Kinshasa province designated for COVID-19 patients care. The study included adult in-patients hospitalized for COVID-19 infection. All patients were confirmed cases of SARS-CoV-2 infection diagnosed by RT-PCR, performed at the National Institute of Bio-medical Research, according to WHO interim guidance [11].

## Site organization

At the start of the epidemic, each of the 25 municipalities of Kinshasa was assigned to different hospitals selected for COVID-19 patients care and designated as COVID-19 treatment center (CTC).

Clinique Ngaliema, a national secondary referral public hospital, located in the municipality of Gombe, was assigned to provide care to COVID-19 patients coming from six municipalities of Kinshasa, including: Gombe, (the municipality currently bearing the highest number of COVID-19 cases), Kintambo, Kinshasa, Lingwala, Barumbu and Ngaliema. These six municipalities represent the catchment area of the Clinique Ngaliema, as a CTC.

Clinique Ngaliema has dedicated a building with a capacity of 27 patients (with possibility to extend to 45 patients) for COVID-19 care and a staff of 107 health professionals. Suspected COVID-19 patients are transferred to the dedicated COVID-19 building, where throat and nasopharyngeal swabs are immediately collected and sent to the National Institute of Biomedical research for RT-PCR analyses.

## Data collection

The following data were collected from patient medical records: demographic information, medical history, exposure history, clinical signs, symptoms and comorbidities, date of symptoms onset, as well as their severity, laboratory examination results, treatment, and disease outcome (death or discharge).

In accordance with to the National guidelines for COVID-19 management in DRC [12], the patients were categorized into mild, moderate and severe clinical groups. The mild clinical group had mild clinical symptoms including: fever, respiratory rate (RR) between 12 and 20 breaths/minute, mild respiratory tract symptoms (rhinorrhea, sore throat and cough), arthralgia, digestives symptoms (diarrhea and vomiting). The moderate clinical group presented: fever, RR between 20 and 30 breaths/ minute, oxygen saturation (SpO2) between 90 and 95%. The severe clinical group presented fever (T˚≥ 38.5˚C), severe respiratory distress, dyspnea, RR ≥ 30 breaths/minute, an oxygen saturation of <90%, Acute respiratory distress syndrome (ARDS), sepsis and septic shock.

Laboratory examination results included: the hemoglobin rate, numbers of leukocytes, percentages of lymphocytes and neutrophils, number of platelets, D-dimer, C-reactive protein (CRP), procalcitonin (PCT), Alanine aminotransferase (ALT) and Aspartate aminotransferase (AST), urea, creatinine, glycaemia, and thick blood smear (TBS) for malaria diagnosis.

## Data analysis

Data were entered and stored in Microsoft Excel 2016. Frequencies, rates and percentages were used to summarize categorical variables, whose proportions were compared using a $X^2$ or Fisher's exact test. Continuous variables were described using the median with interquartile range(IQR). Differences in median values were assessed using the Mann-Whitney test. Values of p less than 0.05 were considered to be statistically significant. Multivariate logistic regression models were constructed to identify factors associated with severity of symptoms, in-hospital death and those associated with unnecessary antimalarial drugs administration. Statistical analyses were done using SPSS statistical program, version 24 (SPSS, Chicago, IL, USA).

## Ethical considerations

The investigators agreed to conduct the present study in full agreement with the principles of the Declaration of Helsinki' and its subsequent relevant amendments. The study was approved

by the institutional ethics board of the Clinique Ngaliema. The access to patient medical records was granted by the director of the hospital. All data were fully anonymized before they have been accessed. Patients whose medical records were selected for analysis sought treatment from march to July 2020 and their data were accessed from Jun to July 2020.

## Results

### Sociodemographic characteristics and seeking care behavior

Between Mars 11[th] to July 22[th] 2020, a total of 160 patients with COVID-19 confirmed infection admitted at Clinique Ngaliema Hospital, and complete data for all variables of interest, were included. The median age of all patients was 54 years (IQR:38–64) (Table 1). The sex ratio was 1.1, with 51% of male (82 cases). The median time from illness onset to hospitalization at Clinique Ngaliema was 7 days (IQR:5–10). Female attended hospital earlier than male (5 days IQR: [4–7] vs 7 days [IQR: 6–11]; p = 0.003, respectively) (Fig 1). Out of all patients admitted, 42 (26%) were transferred from non-CTC health facilities. Eighty-three (52%) cases were located within the Clinique Ngaliema catchment area (Table 1), with Ngaliema municipality having the highest number of cases (51%) (Fig 2). Sixty-eight (43%) patients had an history of exposure to a COVID-19 confirmed or suspected case and most of them (51 cases), were in the mild group (Table 1). In addition, patients with clear history of exposure to COVID case were more likely to attend Clinique Ngaliema as a CTC, earlier (5 days [IQR:

**Table 1. Demographic and baseline characteristics of patients with SARS-CoV-2 infection.**

| | All patients (N = 160) | Mild (n = 92) | Moderate (n = 19) | P value (Moderate vs Mild) | Severe (n = 49) | P value (severe vs Mild) | P value (Severe vs Moderate) |
|---|---|---|---|---|---|---|---|
| **Variables** | | | | | | | |
| Age (year) | 54 (38–64) | 51 (35–61) | 50 (38–66) | 0.52 | 58 (50–70) | 0.001* | 0.081 |
| Sex | - | - | - | 0.52 | - | 0.035* | 0.425 |
| Male | 82 (51%) | 41 (45%) | 10 (53%) | - | 31 (63%) | - | - |
| Female | 78 (49%) | 51 (55%) | 9 (47%) | - | 18 (37%) | - | - |
| Exposure history | | | | | | | |
| Contact with suspected or confirmed COVID 19 cases | 68 (43%) | 51 (55%) | 7 (37%) | 0.155 | 10 (20%) | 0.001* | 0.273 |
| No clear contact history | 7 (4%) | 4 (4%) | 0 | - | 3 (6%) | - | - |
| History of recent travel outside DRC | 5 (3%) | 3 (3%) | 0 | - | 2 (4%) | - | - |
| Comorbidities | | | | | | | |
| Any | 74 (46%) | 31 (33%) | 11 (58%) | 0.048* | 32 (65%) | 0.0003* | 0.572 |
| Hypertension | 55 (34%) | 24 (26%) | 10 (53%) | 0.023* | 21 (43%) | 0.043* | 0.471 |
| Diabetes | 31 (19%) | 14 (15%) | 5 (26%) | 0.244 | 12 (24%) | 0.178 | 0.877 |
| Obesity | 13 (8%) | 7 (8%) | 2 (11%) | - | 4 (8%) | - | - |
| Heart disease | 11 (7%) | 2 (2%) | 2 (11%) | - | 7 (14%) | - | - |
| Asthma / chronic pulmonary disease | 5 (3%) | 1(1%) | 1 (5%) | - | 3 (6%) | - | - |
| Time before seeking care (day) | 7 (5–10) | 5 (4–7) | 7 (4.5–8.5) | 0.817 | 7 (6.5–10) | 0.003 * | 0.074 |
| Therapeutic itinerary | - | - | - | 0.006* | | <0.0001 | 0.589 |
| Transferred from non-CTC hospitals | 42 (26%) | 11 (12%) | 8 (42%) | - | 23 (47%) | - | - |
| Attended Cl.Ng. from home | 118 (74%) | 81 (88%) | 11 (58%) | - | 26 (53%) | - | - |
| Within Cl.Ng. Coachtment area | 83 (52%) | 47 (51%) | 9 (47%) | 0.768 | 27 (55%) | 0.651 | 0.569 |
| Self-reporting Antimalarial treatment before hospitalization | 70 (44%) | 27 (29%) | 12 (63%) | 0.043* | 31 (64%) | 0.0003* | 0.576 |

Data are n (%) or median (IQR), Cl.Ng.: Clinique Ngaliema CTC: COVID-19 treatment center, COVID 19: Coronavirus disease 2019.

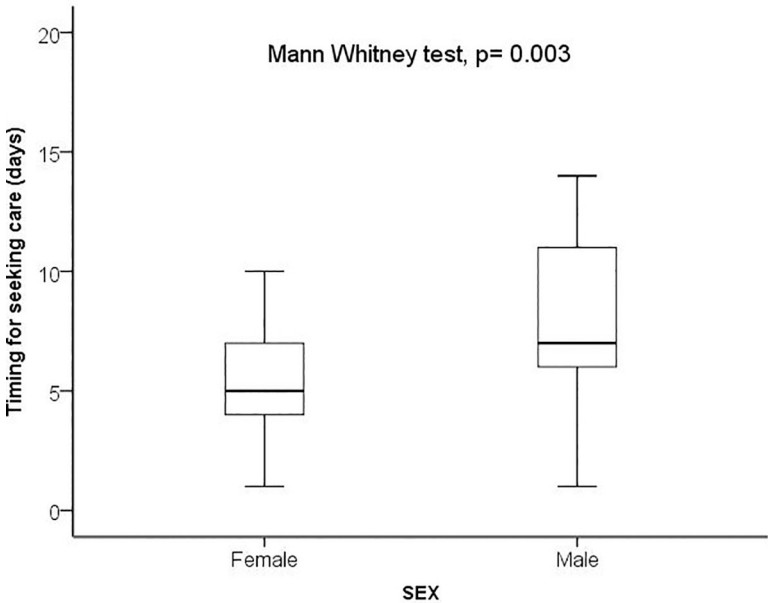

**Fig 1. Sex and time interval from onset of symptoms to hospitalization.**

3–7] from the onset of symptoms) compared to those with unclear history of exposure (7 days [IQR: 5–8]; p = 0.034) (Fig 3).

## Past medical history

Comorbidities were reported among 74 (46%) patients and the most common included hypertension and diabetes. Severe patients were more likely to have comorbidities compared to mild ones (65% vs 33%; p = 0.0003) (Table 1).

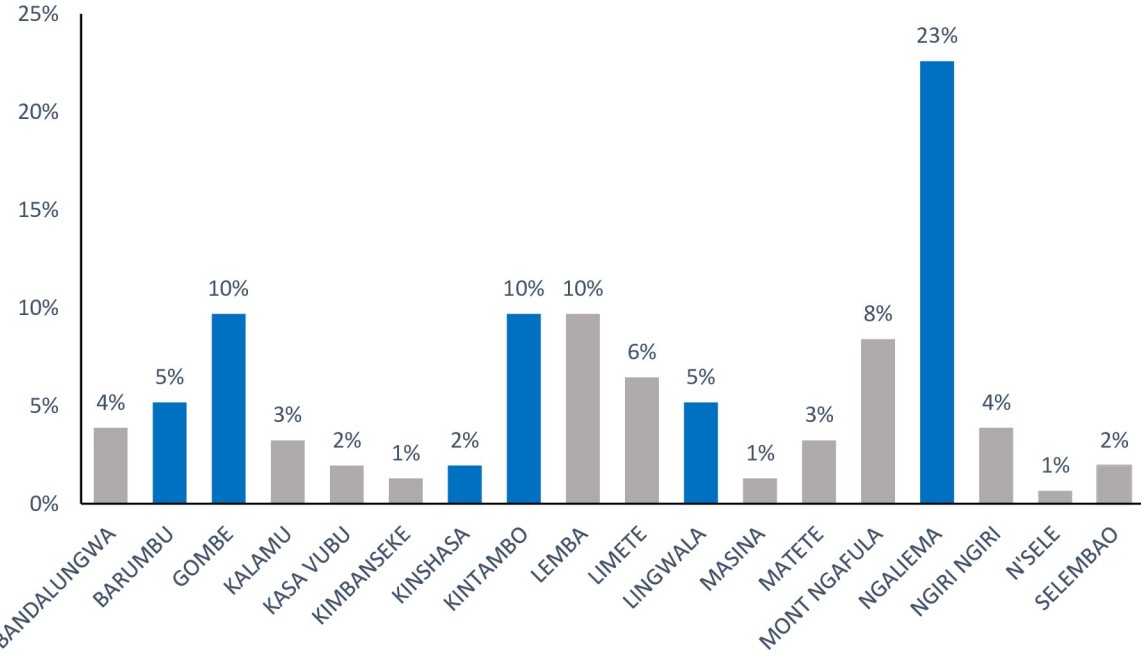

**Fig 2. Distribution of COVID-19 patients according to their location in the province of Kinshasa.** Municipalities within Clinique Ngaliema catchment area are presented in blue and others in grey.

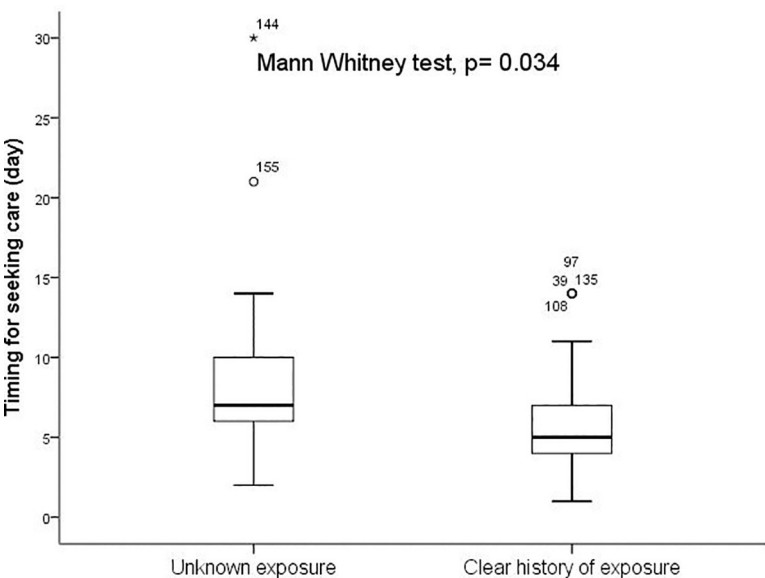

**Fig 3. History of exposure to a confirmed or suspected COVID-19 case and time interval from onset of symptoms to hospitalization.**

### Initial clinical presentation

Patients were clinically categorized into three groups: mild (92 [57%] cases), moderate (19 [12%] cases) and severe (49 [31%] cases). Patients in the severe group were significantly older than those in the mild group (median age 58 years [IQR: 50–70] vs 51 years [IQR: 35–61], p = 0.001) (Table 1).

The most common symptoms reported upon admission were fever (93 [58%] cases), cough (92 [57%] cases), fatigue (87 [54%] cases), and shortness of breath (72 [45%] cases). Vital signs at admission, indicated that 56 (35%) patients had heart rate (HR) >100 beats/min, 18 (11%) presented RR ≥30 breaths/min, and 34 (21.4%) had SpO2 below 90% (Table 2). Female had significantly higher SpO2 compared to male (median SpO2 97% [IQR:88–99] vs 92% [IQR: 82–97]; p = 0.009) (Fig 4). At baseline, QTc interval prolongation was found in 8 (5%) patients (Table 2).

### Hematology and inflammatory markers

Most patients (115 [72.1%] cases) had normal leukocyte count. Severe patients had significantly higher leukocyte count compared to mild and moderate patients. The median CRP concentration was 48 (IQR: 6–115) for all patients and CRP level was 2.5 times higher in severe patients than in mild patients (84 mg/L [IQR: 24–218] vs 34 [IQR: 2–95]; p = 0.001). The procalcitonin level was also elevated (> 0,5 μg/L) in most patients (118 [74%] cases), with no significant difference between clinical groups (Table 3).

### Coagulation function

The level of D-dimer was above 1000 ng/mL in 107 cases (67%) and no significant difference of D-dimer concentration was found between clinical groups (Table 3).

### Liver function

Elevated concentration of AST (>38 IU/L) was observed in was 106 (66%) cases. Compared to mild patients, severe patients had significantly higher AST concentration (p = 0.005). Elevated

**Table 2. Symptoms, vital signs and outcomes of patients infected with SARS-CoV-2 infection.**

| | All patients (N = 160) | Mild (n = 92) | Moderate (n = 19) | P value (Moderate vs Mild) | Severe (n = 49) | P value (severe vs Mild) | P value (Severe vs Moderate) |
|---|---|---|---|---|---|---|---|
| **Variables** | | | | | | | |
| Self-reported symptoms | | | | | | | |
| Fever | 93 (58%) | 39 (42%) | 14 (74%) | 0.013* | 40 (82%) | <0.0001* | 0.470 |
| Cough | 92 (57%) | 45 (49%) | 13 (68%) | 0.041* | 34 (69%) | 0.007* | 0.854 |
| Fatigue | 87 (54%) | 34 (37%) | 12 (63%) | 0.036* | 41 (84%) | <0.0001* | 0.069 |
| Shortness of breath | 72 (45%) | 13 (14%) | 16 (84%) | <0.0001* | 43 (88%) | <0.0001* | 0.701 |
| Myalgia or arthralgia | 33 (21%) | 17 (18%) | 1 (5%) | - | 15 (31%) | - | - |
| Sore throat | 18 (11%) | 9 (10%) | 3 (18%) | - | 6 (12%) | - | - |
| Rhinorrhea | 16 (10%) | 8 (9%) | 2 (11%) | - | 6 (12%) | - | - |
| Vomiting | 12 (8%) | 3 (3%) | 3 (16%) | - | 6 (12%) | - | - |
| Diarrhea | 11 (7%) | 3 (3%) | 2 (11%) | - | 6 (12%) | - | - |
| Headache | 7 (4%) | 6 (7%) | 1 (5%) | - | 0 | - | - |
| Loss of appetite | 5 (3%) | 1 (1%) | 0 | - | 4 (8%) | - | - |
| Olfactory disorders | 2 (1%) | 2 (2%) | 0 | - | 0 | - | - |
| Vital signs upon admission | | | | | | | |
| Respiratory rate (RR) _breaths/min | 22 (20–26) | 22 (20–22) | 24 (22–28) | 0.0006* | 27 (26–30) | <0.0001* | 0.021* |
| *Dyspnea (RR ≥ 30 breaths/ minute)* | 18 (11%) | 5 (5%) | 3 (16%) | - | 10 (20%) | <0.0001* | 0.004* |
| Temperatures _ ˚C | 36.8 (36.5–37.4) | 36.6 (36.4–36.9) | 37.3 (36.8–38.3) | 0.02* | 37.8 (36.5–38.3) | 0.025* | 0.080 |
| *Temperature ≥ 38˚C* | 62 (39%) | 34 (37%) | 9 (47%) | 0.243 | 19 (39%) | 0.203 | 0.698 |
| *SpO₂ _%* | 96 (86–98) | 98 (96–99) | 91 (90–97) | <0.0001* | 80 (62–87) | <0.0001* | <0.0001* |
| *SpO2 < 95%* | 58 (36%) | 5 (5%) | 12 (63%) | <0.0001* | 41 (84%) | <0.0001* | 0.112 |
| Heart rate (HR) _beats/min | 89 (84–104) | 85 (80–92) | 88 (85–112) | 0.042* | 103 (95–116) | <0.0001* | 0.143 |
| *HR>100 beats/min* | 56 (35%) | 13 (14%) | 8 (41%) | 0.012* | 35 (71%) | <0.0001* | 0.115 |
| Systolic blood pressure (SBP) _mmHg | 120 (110–135) | 120 (110–130) | 130 (120–140) | 0.013* | 130 (110–140) | 0.085 | 0.521 |
| SBP <90 mmHg | 3 (2%) | 1(1%) | 0 | - | 2 (4%) | - | - |
| ECG QTc _ms | 432 (407–458) | 424 (404–445) | 497 (434–507) | - | 441 (411–454) | - | - |
| *ECG QTc ≥ 500 ms* | 8 (5%) | 1 (1.1%) | 4 (21%) | - | 3 (6%) | - | - |
| Treatment | | | | | | | |
| Hydroxychloroquine-Azithromycin | 147 (92%) | 88 (96%) | 17 (89%) | 0.280 | 42 (86%) | 0.048* | 0.683 |
| Oxygen support | 81 (51%) | 17 (18%) | 17 (89%) | <0.0001* | 47 (96%) | <0.0001* | 0.325 |
| Antibiotic therapy | 90 (56%) | 33 (36%) | 14 (74%) | 0.003* | 43 (87%) | <0.0001* | 0.184 |
| Dexamethasone | 5 (3%) | - | - | - | 5 (3%) | - | - |
| Invasive ventilation | 4 (2.5%) | - | - | - | 4 (8.2%) | - | - |
| Clinical evolution | | | | | | | |
| Hospitalization duration | 15 (4–20) | 16 (13–21) | 19 (9–23) | 0.731 | 3.5 (1–15) | <0.0001* | 0.0002* |
| Outcome | - | - | - | 0.11 | - | <0.0001* | 0.008* |
| Discharge | 128 (80%) | 87 (95%) | 16 (84%) | - | 25 (51%) | - | - |
| Death | 32 (20%) | 5 (5%) | 3 (16%) | - | 24 (49%) | - | - |

Data are n (%) or median (IQR), SpO2: Peripheral oxygen saturation, SBP: Systolic blood pressure, ECG QTc: Electrocardiogram Corrected QT interval.

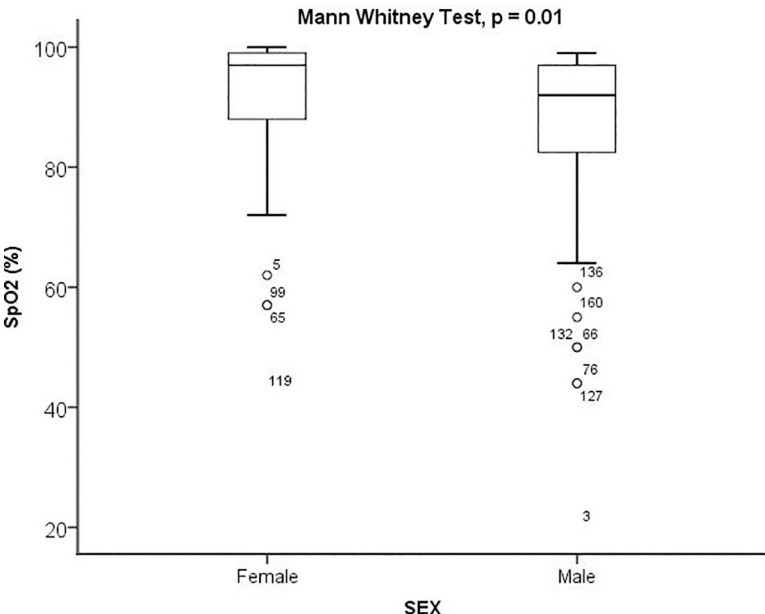

**Fig 4. Sex and peripheral oxygen saturation.**

ALT level (>45 IU/L) was found in 60 (37%) patients, with no significant difference between clinical groups (Table 3).

## Renal function

The median urea level of all the patients was nearly in the normal range (27mg/dL [IQR 17–43]) (Normal range: 15–45 mg /L) and severe patients had higher urea concentration compared to mild ones (p<0.007). The creatinine concentration of all patients was equally nearly in the normal range (9 mg/L [IQR: 7–13]) (normal range: 5–15 mg/L) with no significant difference between clinical groups (Table 3).

## COVID-19 treatment

Almost all patients (147 [92%]) received the combination of hydroxychloroquine (or chloroquine phosphate)—azithromycin (HCQ-AZ). Beside HCQ-AZ, patients also received antibiotic (90 [56%] cases) and corticosteroids (5 [3%] cases) as concomitant or adjuvant treatment, respectively. Many patients (81 [51%] cases) required oxygen support and most of them were in the severe group (Table 2). Invasive ventilation was used for very few patients (4 [2.5%] cases).

## Factors affecting in-hospital death

Of all patients included in the study, 32 (20%) died during hospitalization and 128 (80%) were discharged. In univariate analysis, odds of in-hospital death was higher in patients who did not receive hydroxychloroquine (or chloroquine phosphate) plus Azithromycin, patients with systemic inflammatory response syndrome (SIRS), higher RR, shortness of breath, elevated D-dimer and PCT or higher quick sequential organ failure assessment (qSOFA) score. Age, comorbidity, fever, SpO2, neutrophil and lymphocyte rates, CRP, AST, urea and creatinine were also associated with death (Table 4). The regression model showed that older age, lower

**Table 3. Laboratory findings on admission.**

| | All patients (N = 160) | Mild (n = 92) | Moderate (n = 19) | P value (Moderate vs Mild) | Severe (n = 49) | P value (severe vs Mild) | P value (Severe vs Moderate) |
|---|---|---|---|---|---|---|---|
| **Variables** | | | | | | | |
| Haemoglobin g/dL | 12 (10–13) | 12 (10–12) | 12 (10–13) | 0.823 | 12 (11–13) | 0.459 | 0.872 |
| White blood cell (WBC) count_ cells/ µL, median (IQR) | 6200 (4600–9012) | 5100 (4200–6900) | 6300 (5400–10000) | 0.012* | 9600 (6800–14000) | <0.0001* | 0.175 |
| WBC count: | - | - | - | 0.653 | | <0.0001* | 0.401 |
| < 4000 | 21 (13%) | 16 (17%) | 1 (6%) | - | 4 (9%) | - | - |
| 4000–12000 | 115 (72%) | 74 (80%) | 14 (75%) | - | 27 (56%) | - | - |
| >12000 | 24 (15%) | 3 (3%) | 4 (19%) | - | 17 (35%) | - | - |
| Neutrophils _%, median (IQR) | 66 (52–78) | 56 (48–66) | 72 (63–75) | 0.020* | 77 (70–85) | 0.0001* | 0.393 |
| Neutrophils_%: | - | - | - | 0.048* | | <0.0001* | 0.701 |
| < 55% | 50 (31%) | 42 (45%) | 3 (14%) | - | 5 (10%) | - | - |
| 55–70% | 43 (27%) | 29 (32%) | 5 (29%) | - | 9 (18%) | - | - |
| >70% | 67 (42%) | 21 (23%) | 11 (57%) | - | 35 (71%) | - | - |
| Lymphocytes _%, median (IQR) | 29 (18–41) | 36 (26–44) | 25 (17–31) | 0.003* | 20 (13–26) | <0.0001* | 0.259 |
| Lymphocytes_%: | - | - | - | 0.030* | | 0.0001* | 0.250 |
| <20% | 44 (28%) | 14 (15%) | 6 (32%) | - | 23 (47%) | - | - |
| 20–40% | 73 (46%) | 43 (47%) | 12 (62%) | - | 19 (39%) | - | - |
| >40% | 43 (27%) | 35 (38%) | 1 (6%) | - | 7 (14%) | - | - |
| Platelet count _ cells per µL, median (IQR) | 204,000 (127,000–285,000) | 199,000 (133,500–281,012) | 199,000 (117,000–314,000) | 0.798 | 209,000 (126,000–311,000) | 0.653 | 0.9872 |
| Platelet count: | - | - | - | 0.024* | | 0.132 | 0.332 |
| < 150,000 | 55 (34%) | 29 (32%) | 9 (47%) | - | 17 (34%) | - | - |
| 150,000–400,000 | 94 (59%) | 60 (65%) | 6 (31%) | - | 28 (57%) | - | - |
| >400,000 | 11 (7%) | 3 (3%) | 4 (21%) | - | 4 (9%) | - | - |
| AST_IU/L, median (IQR) | 69 (30–115) | 55 (21–105) | 69 (26–116) | 0.531 | 92 (54–136) | 0.005* | 0.145 |
| AST >38 | 106 (66%) | 55 (60%) | 13 (68%) | 0.416 | 38 (78%) | 0.016* | 0.363 |
| ALT_IU/L, median (IQR) | 37 (18–63) | 32 (16–64) | 44 (18–58) | 0.802 | 39 (25–67) | 0.338 | 0.822 |
| ALT >45 | 60 (37%) | 33 (36%) | 9 (47%) | 0.407 | 18 (36%) | 0.721 | 0.597 |
| CRP_ mg/L, median (IQR) | 48 (6–115) | 34 (2–95) | 56 (12–96) | 0.333 | 84 (24–218) | 0.001* | 0.186 |
| PCT _ µg/L, median (IQR) | 22 (0.42–55) | 26 (0.2–54) | 30 (0.17–67) | 0.506 | 13 (1.3–60) | 0.453 | 0.647 |
| PCT >0.5 | 118 (74%) | 64 (70%) | 13 (64%) | 0.6 | 41 (84%) | 0.082 | 0.083 |
| D-dimer_ ng/mL, median (IQR) | 2390 (613–5483) | 1549 (406–5127) | 4143 (919–6312) | 0.157 | 3582 (1256–4814) | 0.066 | 0.781 |
| D-dimer_mg/mL: | - | - | - | 0.343 | - | 0.005* | 0.224 |
| <500 | 32 (20%) | 28 (30%) | 2 (13%) | - | 2 (5%) | - | - |
| 500–1000 | 21 (13%) | 11 (12%) | 4 (19%) | - | 6 (12%) | - | - |
| >1000 | 107 (67%) | 53 (58%) | 13 (68%) | - | 41 (83%) | - | - |
| Urea_ mg/dL, median (IQR) | 27 (17–45) | 24 (16–38) | 24 (22–38) | 0.985 | 39 (18–75) | 0.007* | 0.055 |
| Urea >45 | 40 (25%) | 16 (17%) | 2 (12%) | 0.213 | 22 (45%) | 0.0008* | 0.015* |
| Creatinine_ mg/L, median (IQR) | 9 (7–13) | 9 (7–12) | 9 (8–12) | 0.703 | 12 (7–17) | 0.088 | 0.354 |
| Creatinine > 15 mg/L | 15 (9%) | 7 (8%) | 0 | - | 8 (16%) | 0.004* | - |
| Glycaemia _g/dL | 123 (90–220) | 108 (86–181) | 137 (99–245) | 0.214 | 156 (104–225) | 0.009* | 0.748 |
| Positive TBS | 1(0.63%) | 0 | 0 | - | 1 (2%) | - | - |
| Malaria parasite density /µL | 16,900 | - | - | - | - | - | - |

Data are n (%) or median (IQR), %: Proportion, ALT: Alanine aminotransferase, AST: Aspartate aminotransferase, CRP: C-reactive protein, PCT: Procalcitonin, TBS: Thick blood smear.

**Table 4. Factors associated with in-hospital death.**

| Variables | | Non-survivor (n = 33) | Survivor (127) | Univariate OR | p value | Multivariate OR | P value |
|---|---|---|---|---|---|---|---|
| Age_years | | 59 (51–69) | 52 (36–62) | 1.04 (1.01–1.07) | 0.006* | 1.06 (1.0–1.11) | 0.033* |
| Sex | Male | 17 (53%) | 66 (52%) | 1.06 (0.49–2.32) | 0.881 | - | - |
| | Female | 16 (47%) | 61 (48%) | 1 | | | |
| Presence of Comorbidity | Yes | 23 (69%) | 53 (42%) | 3.06 (1.34–7.02) | 0.008* | - | - |
| | No | 10 (31%) | 74 (58%) | 1 | | | |
| Hypertension | Yes | 15 (44%) | 42 (33%) | 1.59 (0.72–3.53) | 0.249 | - | - |
| | No | 18 (56%) | 85 (67%) | 1 | | | |
| Diabetes | Yes | 9 (28%) | 22 (17%) | 1.88 (0.76–4.65) | 0.168 | - | - |
| | No | 24 (72%) | 105 (83%) | 1 | | | |
| Obesity | Yes | 3 (9%) | 9 (7%) | 1.30 (0.33–5.11) | 0.708 | - | |
| | No | 30 (91%) | 118 (93%) | 1 | | | |
| Heart disease | Yes | 4 (12%) | 8 (6%) | 2.35 (0.64–8.58) | 0.197 | - | - |
| | No | 29 (88%) | 119 (94%) | 1 | | | |
| Respiratory rate_breaths/ min | ≤24 | 8 (24%) | 98 (77%) | 10.71 (4.04–28.36) | <0.0001* | - | - |
| | >24 | 25 (76%) | 29 (23%) | 1 | | | |
| SpO2_%, median (IQR) | | 81 (62–87) | 97 (91–99) | 0.92 (0.89–0.96) | <0.0001* | 0.94 (0.90–0.98) | 0.007* |
| Fever | Yes | 26 (78%) | 67 (53%) | 3.13 (1.26–7.78) | 0.014* | - | - |
| | No | 7 (22%) | 60 (47%) | 1 | | | |
| Cough | Yes | 22 (67%) | 70 (55%) | 1.96 (0.83–4.58) | 0.12 | - | - |
| | No | 11 (33%) | 57 (45%) | 1 | | | |
| qSOFA | | 1.0 (1.0–2.0) | 1.0 (0.0–1.0) | 5.11 (2.21–11.85) | 0.0001* | 4.02 (0.84–19.22) | 0.081 |
| SIRS | Yes | 32 (96%) | 77 (61%) | 15.57 (2.03–119.27) | 0.008* | - | - |
| | No | 1 (4%) | 50 (39%) | 1 | | | |
| Shortness of breath | Yes | 27 (81%) | 48 (38%) | 7.15 (2.74–18.68) | 0.0001* | - | - |
| | No | 6 (19%) | 79 (62%) | 1 | | | |
| Hydroxychloroquine (or Chloroquine phosphate)- Azithromycin | Yes | 25 (75%) | 122 (96%) | 0.13 (0.04–0.43) | 0.0008* | 0.24 (0.03–2.2) | 0.208 |
| | No | (25%) | 5 (4%) | 1 | | | |
| Therapeutic itinerary | Attended directly Cl. Ng. | 19 (58%) | 97 (76%) | 2.24 (0.98–5.13) | 0.056 | - | - |
| | Transferred from non-CTC | 14 (42%) | 30 (24%) | 1 | | | |
| Heart rate_beats /min | | 111 (98–119) | 88 (82–99) | 1.07 (1.04–1.11) | <0.0001* | 1.06 (1.02–1.11) | 0.027* |
| WBC counts_cells/µL, median (IQR) | | 8700 (6300–13300) | 5800 (4500–8400) | 1 (1–1) | 0.105 | - | - |
| Neutrophils_%, median (IQR) | | 75 (69–85) | 62 (50–75) | 1.05 (1.02–1.09) | 0.004* | - | - |
| Lymphocytes_%, median (IQR) | | 21 (15–28) | 31 (19–42) | 0.96 (0.93–0.99) | 0.024* | | |
| PCT_ µg/L, median (IQR) | <0.5 | 3 (8%) | 41 (32%) | 1 | | | |
| | ≥0.5 | 30 (92%) | 86 (68%) | 5.18 (1.16–23.26) | 0.032* | - | - |
| CRP_ mg/L, median (IQR) | | 148 (27–319) | 42 (0.5–96) | 1.01 (1–1.01) | 0.0002* | - | - |
| D-dimer_ mg/L | <1000 | 3 (8%) | 47 (37%) | 1 | | | |
| | ≥1000 | 30 (92%) | 80 (63%) | 6.33 (1.41–28.42) | 0.016* | - | - |

*(Continued)*

**Table 4.** (Continued)

| Variables | | Non-survivor (n = 33) | Survivor (127) | Univariate OR | p value | Multivariate OR | P value |
|---|---|---|---|---|---|---|---|
| ALT_UI/L | | | 32 (15–59) | 55 (29–77) | 1.01 (0.99–1.02) | 0.062 | - | - |
| AST_ UI/L | | | 22 (60–102) | 83 (128–159) | 1.02 (1.01–1.03) | <0.0001* | 1.02 (1.01–1.03) | 0.005* |
| Urea_ mg/dL | | | 35 (20–73) | 24 (17–42) | 1.01 (1–1.03) | 0.015* | | |
| Creatinine_ mg/dL | ≤15 | 26 (79%) | 119 (94%) | 1 | | | |
| | >15 | 7 (21%) | 8 (6%) | 3.95 (1.14–13.75) | 0.031* | - | - |

Data are n (%) or median (IQR), %: Proportion, ALT: Alanine aminotransferase, AST: Aspartate aminotransferase, CRP: C-reactive protein, PCT: Procalcitonin, SpO2: Peripheral oxygen saturation, qSOFA = Quick Sequential Organ Failure Assessment, SIRS: Systemic Inflammatory Response Syndrome, OR: Odds ratio.

SpO2, higher heart rate and elevated AST at admission were independent risk factors associated with in-hospital death (Table 4).

## COVID-19 and malaria co-infection

Only one case (0.63%) of COVID-19 and malaria co-infection was observed. The parasite density was 16,900/μL (Table 3). Antimalarial treatment before hospitalization was reported by 70 (44%) cases and more than half (61%) of them were found in the moderate and severe groups (Table 1). Factors associated with antimalarial use are presented in the appendix (see web-only S1 Table).

## Discussion

This retrospective study portrayed the clinical features of COVID-19 patients hospitalized at Clinique Ngaliema and living in Kinshasa, in the DRC.

The median age was found to be 54 years and severe patients were older than mild patients. Previous studies reported a broadly similar age distribution [13, 14]. Male and female were equally represented. Equal sex distribution was also reported by Suxin Wan et al [15], although many studies in contrast reported gender imbalance, with either higher male to female ratio [5, 16] or higher female to male ratio [4]. Interestingly, in line with other reports [17], it was noted that female were less likely to have severe symptoms compared to male. However, our results suggested that female seemed to be less affected by the infection because they attended hospital at its early stage, characterized by mild symptoms [18], compared to male who reached hospital at the progressive stage of the disease [18]. In addition, to support this observation, this study did not find any association between gender and in-hospital death.

Of all patients, 43% had a clear history of exposure to a confirmed or probable COVID-19 case. These patients were prone to visited directly from home a CTC, and at an earlier stage of the illness, compared to those with unclear history of exposure. These findings could suggest a higher level of awareness about COVID-19 existence and related symptoms in patients living in the city center of Kinshasa, who visited Clinique Ngaliema. Therefore, efforts should be invested in order to increase awareness in peri-urban areas of Kinshasa and in other provinces of DRC affected by COVID-19, where awareness on COVID-19 disease is expected to be lower.

Almost half of all patients (44%) reported having received antimalarial treatment before their hospitalization at the Clinique Ngaliema. However, it was unclear whether they have been diagnosed for malaria or they received antimalarial drugs on the basis of underlying fever

or other malaria related symptoms. As malaria and COVID-19 share a couple of symptoms, a number of mild COVID-19 cases may go undetected and treated as *Plasmodium spp* infections. These findings underline the complexity of the of COVID-19 infection management in malaria endemic areas. Next to that, in line with other reports [14], more than half of patients received antibiotic therapy. These findings are a matter of great concern and underline the urgent need for strong guidelines in order to reduce and prevent antibiotic and antimalarial overuse during COVID-19 pandemic, in sub-Saharan Africa.

Comorbidity was reported by almost half (46%) of all patients and the two most common were hypertension, and diabetes. This agrees with other studies [5]. No significant difference of comorbidity was observed between sex. In line with previous studies [19], patients with any underlying comorbidity were more likely to have severe symptoms and among reported comorbidities, hypertension was more frequently observed in severe patients. However, in contrast with other findings suggesting that comorbidities, mainly pre-existing cardiovascular disease could be associated with COVID-19 mortality [19], hypertension and heart disease were not associated with in-hospital death.

Fever, cough and fatigue were the top three common symptoms reported upon admission. These symptoms were also reported following the same order, by previous reports [13]. This list of common symptoms may help medical staff to recognise suspected COVID-19 cases, and encourage self-isolation or hospitalization, especially in remote areas with very poor COVID-19 detection capacities. Nevertheless, other non-common symptoms deserve similar attention than the more common ones.

Severe patients had significantly higher leucocyte count, elevated neutrophil rate, and lower lymphocyte rate, compared to other clinical groups. This has been also documented elsewhere [20]. Our study also revealed that most patients had elevated PCT and CRP, and higher CRP concentration was also associated with the disease severity. These inflammatory abnormalities were also reported previously [21].

Coagulation disorders, measured by the level of D-dimer were observed in more than half patients (67%). Elevated D-dimer was also reported in previous studies [22]. In line with other previous findings [23], our study also revealed that elevated D-dimer was associated with the disease severity.

Liver enzymes abnormalities were found in 66% of patients. This higher proportion of liver dysfunction on admission could be explained, by the observed time interval ($\geq$7 days) from initial symptoms to hospitalization, which coincides with the time of occurrence of severe symptoms [13]. High proportions of AST and ALT elevation were also reported in other studies [24]. Previous studies have also shown that abnormalities of liver function, especially elevation of AST were significantly associated with COVID-19 severity [25] and mortality [26]. Our study also documented that severe group had the highest concentration of AST.

We observed a hospital case fatality rate of 20%, which seemed to be lower than those reported by other studies [14]. However, this is unlikely to reflect the true fatality rate of the disease, as out-patients and those with missing data were excluded. Independent risk factors associated with in-hospital death in this study included older age, lower SpO2, higher heart rate and higher AST level. Association of age and death [14] as well as lower SpO2 and death was also reported in China [27]. Oxygen saturation has appeared to be a reliable surrogate marker of COVID-19 illness severity and death. However, to the best of our knowledge, heart rate was only described as a predictor for a positive SARS-CoV-2 test [17].

This study reported a low prevalence (0.63%) of malaria and COVID-19. A couple of reasons may support this observation. First, the lower prevalence of malaria observed in the city center (the catchment area of Clinique Ngaliema and its neighboring communities) compared to peri-urban areas [28]; and second, the considerable proportion of patients who received

antimalarial drugs before hospitalization. This is to our knowledge the first study reporting the coexistence of malaria and COVID-19 infections. Nevertheless, previous studies also have shown that co-infections including other viral, bacterial and fungal infections were rarely found among patients with COVID-19 [29].

The study has a number of limitations. First, lack of some key data (laboratory tests) and excluded variables from analysis because of missing values (high sensitive troponin, ferritin, blood gas, ionogramm etc.) that could play an important role, due to the retrospective design. Second, excluded cases, could lead to a selection bias. However, this is unlikely given that patients excluded were similar to those included in the study, in regard to all socio-demo-graphic data. Last, although nearly half of patients included were located out of the catchment area of Clinique Ngaliema, our sample may not be representative of Kinshasa. Therefore, our findings should be generalized to COVID patients of Kinshasa with caution.

## Supporting information

**S1 Table. Factors associated with antimalarial use in COVID-19 patients.**
(DOCX)

## Acknowledgments

The authors thank all other health care workers of Clinique Ngaliema COVID-19 treatment center involved in the diagnosis and treatment of patients. We acknowledge Mr Damas Mas-solo and Mrs Ida Dikebelayi, Dr Freddy Kibwana, Dr Thierry Mbazu, for assisting in data collection, as well as Mr Matthieu Tshitamba, Mr Fefe Baleka, Mrs Bibi Bwiri and Mr Toguy Ndombe for assisting in sample collection and laboratory analysis.

## Author Contributions

**Conceptualization:** Junior Rika Matangila.

**Data curation:** Junior Rika Matangila, Ritha Kibambe Nyembu, Gloire Mosala Telo, Christian Djuba Ngoy, Taty Msueun Sakobo, Joel Mbembo Massolo, Blandine Mbo Muyembe.

**Formal analysis:** Junior Rika Matangila.

**Methodology:** Junior Rika Matangila.

**Validation:** Ritha Kibambe Nyembu.

**Writing – original draft:** Junior Rika Matangila.

**Writing – review & editing:** Ritha Kibambe Nyembu, Gloire Mosala Telo, Christian Djuba Ngoy, Taty Msueun Sakobo, Joel Mbembo Massolo, Blandine Mbo Muyembe, Richard Kapela Mvwala, Cedric Kabemba Ilunga, Emmanuel Bakilo Limbole, Jeff Mukengeshay Ntalaja, Roger Minga Kongo.

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
