## [Decision Letter · Decision Letter 0]

9 Oct 2020

PONE-D-20-28772

Clinical characteristics of COVID-19 patients hospitalized at Clinique Ngaliema, a Public hospital in Kinshasa, in the Democratic Republic of Congo: a retrospective cohort study

PLOS ONE

Dear Dr. Matangila,

Thank you for submitting your manuscript to PLOS ONE. After careful consideration, we feel that it has merit but does not fully meet PLOS ONE’s publication criteria as it currently stands. Therefore, we invite you to submit a revised version of the manuscript that addresses the points raised during the review process.

The reviewers have commented on your above paper. They have suggested that this manuscript be revised according to the reviewers suggestions and resubmitted.  Provided you address the changes recommended, the manuscript will be accepted for publication. 

We look forward to receiving your revised manuscript.

Kind regards,

Prof. Raffaele Serra, M.D., Ph.D

Academic Editor

PLOS ONE

Journal Requirements:

2. In the ethics statement in the manuscript and in the online submission form, please provide additional information about the patient records used in your retrospective study, including: a) whether all data were fully anonymized before you accessed them; b) the date range (month and year) during which patients' medical records were accessed; c) the date range (month and year) during which patients whose medical records were selected for this study sought treatment. If patients provided informed written consent to have data from their medical records used in research, please include this information.

4.We note that you have indicated that data from this study are available upon request. PLOS only allows data to be available upon request if there are legal or ethical restrictions on sharing data publicly. For information on unacceptable data access restrictions, please see http://journals.plos.org/plosone/s/data-availability#loc-unacceptable-data-access-restrictions.

Additional Editor Comments (if provided):

The reviewers have commented on your above paper. They have suggested that this manuscript be revised according to the reviewers suggestions and resubmitted.

Reviewers' comments:

Reviewer's Responses to Questions

**Comments to the Author**

1. Is the manuscript technically sound, and do the data support the conclusions?

Reviewer #1: Yes

2. Has the statistical analysis been performed appropriately and rigorously? 

Reviewer #1: Yes

3. Have the authors made all data underlying the findings in their manuscript fully available?

Reviewer #1: Yes

4. Is the manuscript presented in an intelligible fashion and written in standard English?

Reviewer #1: Yes

5. Review Comments to the Author

Reviewer #1: This authors describe the epidemiological and clinical features, laboratory findings, treatment, and

outcomes of COVID‐ 19 patients in one of the selected Hospital for COVID-19 care of Kinshasa, in Democratic Republic of Congo.

The study is very interesting and novel. I wonder: Regarding prognosis what was the relationship with underlying cardiovascular disease in your study? Possibly you may focus on this issue in the discussion section. For example citing also this recent article by Ielapi N et al. Cardiovascular disease as a biomarker for an increased risk of COVID-19 infection and related poor prognosis. Biomark Med. 2020;14(9):713-716, which deals with a poor prognosis for cardiovascular patients.

6. PLOS authors have the option to publish the peer review history of their article (what does this mean?). If published, this will include your full peer review and any attached files.

Reviewer #1: No

---

## [Author Response · Author response to Decision Letter 0]

6 Dec 2020

Journal Requirements:

R/ OK 

 2. In the ethics statement in the manuscript and in the online submission form, please provide additional information about the patient records used in your retrospective study, including: a) whether all data were fully anonymized before you accessed them; b) the date range (month and year) during which patients' medical records were accessed; c) the date range (month and year) during which patients whose medical records were selected for this study sought treatment. If patients provided informed written consent to have data from their medical records used in research, please include this information.

R/ The following additional information have been added: 

• All data were fully anonymized before they have been accessed. Patients whose medical records were selected for analysis sought treatment from March to July 2020 and their data were accessed from Jun to July 2020. 

R/ Patients did not sign a written informed consent to have their medical data used in research. However, upon admission, patients or their legal surrogate signed a hospitalized patient’s charter that specified that their medical records could be further used for research. 

R/ My ORCID iD is : 0000-0002-9025-3604

4.We note that you have indicated that data from this study are available upon request. PLOS only allows data to be available upon request if there are legal or ethical restrictions on sharing data publicly. For information on unacceptable data access restrictions, please see http://journals.plos.org/plosone/s/data-availability#loc-unacceptable-data-access-restrictions.

 R/ The data is the property of the Clinique Ngaliema Hospital, and the latter has imposed a restriction on the availability of the data because, even if they were anonymized, taking into account the individually identifiable health information, in particular: name, full address, date of birth and professional affiliation, which have been deleted, they still contain information such as: age, sex, community or municipality, past medical history of the patients and the period they were hospitalized. These information combined, still have the potential to be linked to the identity of the patient to whom it corresponds, by a patient family member or a close person. In the Democratic Republic of Congo, after surviving COVID-19 infection, a number of persons face stigma in their communities and work places. This is nowadays a matter of great concern. Therefore, discovering their data online, in the format of a dataset, may drive patients to become reluctant to sick care at Clinique Ngaliema. In addition, these data also contain sensitive patient information such as death and abnormal laboratory results. For these reason, the hospital would prefer to share data for research purposes, upon reasonable request, accompanied by a research proposal with a detailed description of the objectives of the study, sent directly to the steering committee of Clinique Ngaliema Hospital at the following e-mail address: infor@cliniquengaliema.org or to the corresponding author: matangilaj@yahoo.fr . This restriction applies only to COVID-19 data. 

 R/ Please consider the answer above 

 Additional Editor Comments (if provided):

The reviewers have commented on your above paper. They have suggested that this manuscript be revised according to the reviewers suggestions and resubmitted.

Reviewers' comments:

Reviewer's Responses to Questions

Comments to the Author

1. Is the manuscript technically sound, and do the data support the conclusions?

Reviewer #1: Yes

2. Has the statistical analysis been performed appropriately and rigorously?

Reviewer #1: Yes

3. Have the authors made all data underlying the findings in their manuscript fully available?

Reviewer #1: Yes

4. Is the manuscript presented in an intelligible fashion and written in standard English?

Reviewer #1: Yes

5. Review Comments to the Author

Reviewer #1: This authors describe the epidemiological and clinical features, laboratory findings, treatment, and

outcomes of COVID‐ 19 patients in one of the selected Hospital for COVID-19 care of Kinshasa, in Democratic Republic of Congo.

The study is very interesting and novel. I wonder: Regarding prognosis what was the relationship with underlying cardiovascular disease in your study? Possibly you may focus on this issue in the discussion section. For example citing also this recent article by Ielapi N et al. Cardiovascular disease as a biomarker for an increased risk of COVID-19 infection and related poor prognosis. Biomark Med. 2020;14(9):713-716, which deals with a poor prognosis for cardiovascular patients.

R/ The following paragraph has been added in the discussion section:

In line with previous studies, patients with any underlying comorbidity were more likely to have severe symptoms and among reported comorbidities, hypertension was more frequently observed in severe patients. However, in contrast with other findings suggesting that comorbidities, mainly pre-existing cardiovascular disease could be associated with COVID-19 mortality, hypertension and heart disease were not associated with in-hospital death. 

And the following article has been cited: Ielapi N et al. Cardiovascular disease as a biomarker for an increased risk of COVID-19 infection and related poor prognosis. Biomark Med. 2020;14(9):713-716, which deals with a poor prognosis for cardiovascular patients.

6. PLOS authors have the option to publish the peer review history of their article (what does this mean?). If published, this will include your full peer review and any attached files.

Do you want your identity to be public for this peer review? For information about this choice, including consent withdrawal, please see our Privacy Policy.

Reviewer #1: No

R/ No attachment found 

---

## [Editor Report · Decision Letter 1]

8 Dec 2020

Clinical characteristics of COVID-19 patients hospitalized at Clinique Ngaliema, a Public hospital in Kinshasa, in the Democratic Republic of Congo: a retrospective cohort study

PONE-D-20-28772R1

Dear Dr. Matangila,

We’re pleased to inform you that your manuscript has been judged scientifically suitable for publication and will be formally accepted for publication once it meets all outstanding technical requirements.

Kind regards,

Prof. Raffaele Serra, M.D., Ph.D

Academic Editor

PLOS ONE

Additional Editor Comments (optional):

amended manuscript is acceptable.
---

## [Editor Report · Acceptance letter]

11 Dec 2020

PONE-D-20-28772R1 

Clinical characteristics of COVID-19 patients hospitalized at Clinique Ngaliema, a Public hospital in Kinshasa, in the Democratic Republic of Congo: a retrospective cohort study   

Dear Dr. Matangila:

I'm pleased to inform you that your manuscript has been deemed suitable for publication in PLOS ONE. Congratulations! Your manuscript is now with our production department. 

Kind regards, 

on behalf of

Prof. Raffaele Serra 

Academic Editor

PLOS ONE